# Developing and validating modular surveys for vector-borne diseases: A study protocol

Skyler Finucane[1,2]☯, Alexandria Renault[2], Mary H. Hayden[3], Kacey Ernst[2]*, Sarah Yeo[4]☯

1 Department of Entomology and Insect Science, University of Arizona, Tucson, Arizona, United States of America, 2 Mel and Enid Zuckerman College of Public Health, University of Arizona, Tucson, Arizona, United States of America, 3 Lyda Hill Institute for Human Resilience, University of Colorado, Colorado Springs, United States of America, 4 The University of Arizona Cancer Center, Tucson, Arizona, United States of America

☯ These authors contributed equally to this work.
* kernst@arizona.edu

## Abstract

Vector-borne diseases are an increasing threat to human health and well-being in the United States. Understanding public perception and practices to reduce vector abundance and vector-human contact can guide effective interventions. Nevertheless, vector-borne disease surveys, which are widely used in the field to understand public perception and practices, are often inconsistent in terms of structure and implementation. This protocol is designed to provide guidance for public health professionals and researchers in the development of future knowledge, attitudes, and practices studies by ensuring uniformity in design and structure. This manuscript describes a rigorous three-phase protocol for the development of standardized vector-borne disease survey modules that can be used throughout the United States to generate data that are comparable across diverse regions. During phase one, a workshop with subject matter experts and a comprehensive literature review will be conducted to identify survey domains and generate items of interest. Survey items will also be mapped based on two theoretical frameworks: the Health Belief Model and the Risks, Attitudes, Norms, Abilities, and Self-Regulation framework. Standards across knowledge, attitudes, and practices surveys will enhance the analysis and interpretation of the data across geographies and time. During phase two, a group of expert judges will evaluate survey items based on content relevance, representativeness, and technical quality. During the final phase, cognitive interviews and surveys with target audience groups will be conducted to measure and ensure the face validity, reliability, and external validity of the modules. Participants will be drawn from a diverse range of educational backgrounds and geographic locations. The surveys developed through this protocol will facilitate acquisition of insights into the public's knowledge, attitudes, and practices concerning vector-borne diseases, allowing for the collection of comparable data across various regions in the United States.

**Data availability statement:** De-identified research data will be made publicly available when the study is completed and published.

**Funding:** This research was generated from the Pacific Southwest Center of Excellence in Vector-Borne Diseases, which has been funded by the cooperative agreement [U01CK000649] from the Centers for Disease Control and Prevention. The contents are solely the responsibility of the authors and do not necessarily represent the official views of the Centers for Disease Control and Prevention. Sarah Yeo (SE) was also supported in part by the National Institutes of Health Training Grant T32CA272303. The funders had no role in study design, data collection and analysis, decision to publish, or preparation of the manuscript.

**Competing interests:** The authors have declared that no competing interests exist.

## Introduction

Vector-borne diseases (VBDs) are an increasing threat to human health and well-being in the United States (US). Over the past two decades, transmission of tick-borne diseases has doubled, while outbreaks of mosquito-borne diseases persist across the continental US and its territories. This increase is associated with shifting seasonal patterns of transmission and the wider spread of disease vectors and their pathogens [1,2]. Engaging the public in prevention and control measures is crucial to mitigating these risks. Understanding public perception and practices taken to reduce vector abundance and vector-human contact can guide effective interventions.

Knowledge, attitudes, and practices (KAP) surveys are one of the most common assessment tools used to understand public health issues in a community [3]. These surveys have been widely used within the context of VBDs. Knowledge questions seek to understand respondents' level of understanding about a VBD topic [3,4]. This could include questions related to knowledge of vectors and their presence, disease transmission and symptoms, and prevention strategies including vector control and personal protection. Attitude questions relate to the emotional, motivational, perceptive, and cognitive beliefs that influence the behaviors of an individual [4]. For VBDs, these questions could pertain to people's attitudes about recommended practices and guidelines regarding the disease, and their perception of risk and disease severity. Practice questions ask about the current actions of an individual, such as the measures undertaken to reduce VBD risk at individual, household, and community levels [4]. In general, KAP surveys are easy to design, quick to implement, easy to interpret, produce quantifiable data, and often allow for generalizations of small sample results to larger populations [5]. It is increasingly recognized that standardized tools to collect data are needed to allow rigorous comparison across different contexts. These endeavors are still relatively nascent in vector-borne disease research. Ideally, KAP surveys use an underlying theoretical framework of behavioral theory, such as the Health Belief Model or the Risk, Attitudes, Norms, Abilities, and Self-Regulation framework, to help gauge individual perceptions of risk severity and self-efficacy. These underlying constructs are important to understand where efforts must be focused to enhance the adoption of prevention and control [6,7]. The use of theoretical frameworks strengthens the rigor, focus, and relevance of the KAP research and captures important domains related to health action and behavior change. Standards across KAP surveys will enhance the analysis and interpretation of the data across geographies and time. These studies set baseline values for subsequent evaluations, evaluate the efficacy of health education initiatives, and inform localized and culturally contextualized intervention approaches [8].

This study aims to create standardized modular surveys for the general public, enabling data comparison across various geographical regions and projects undertaken by public health professionals and researchers in the US. Three phases will be implemented to achieve this goal: 1) identifying survey domains and items, 2) developing modules, and 3) evaluating modules. This manuscript

outlines the detailed protocol to create these modular surveys, which will cover the most prevalent VBDs found in the US as follows:

- General mosquito-borne disease module
- Specific mosquito-borne disease modules
  - Dengue
  - Zika
  - West Nile
  - St. Louis Encephalitis
- General tick-borne disease module
- Specific tick-borne disease modules
  - Lyme disease
  - Rocky Mountain Spotted Fever

## Methods

### Study design

Modular survey development is based on the KAP model [3,4]. The development process consists of three phases, following the methodologies delineated by Hinkin and Boateng (Fig 1) [9,10]. Ethical approval was obtained through the University of Arizona Institutional Review Board in February 2025.

### Phase one: Domain identification and item generation

In this initial phase, a workshop with subject matter experts and a comprehensive literature review will be conducted to identify survey items used in previous studies. The workshop and literature review will be implemented to identify domains and generate items of interest.

**Fig 1. An overview of the three protocol phases.**

**Solicitation of expert panel for key domains.** An initial workshop will be conducted to identify primary domains of KAP that are considered necessary for the control and prevention of mosquito-borne and tick-borne diseases. Approximately 20–30 experts will be drawn from applied public health, vector control agencies, and academic researchers conducting applied research related to VBDs. An asynchronous method, Idea Flip, will be used to solicit initial key domains. A follow-up workshop will bring participants together to identify additional themes and prioritize themes for tick-borne and mosquito-borne diseases.

**Literature review and assessment of existing scales.** The goal of the literature review is to identify and analyze existing surveys and develop a set of domains and items to be used throughout the modular surveys. This is the first step in developing effective measurement tools and ensuring content validity [9]. The literature review will be conducted in several stages. Initially, a comprehensive literature review will be undertaken using specific search strategies. Search criteria will include "KAP," "KAB," or "KAPP" combined with "AND" followed by specific mosquito and tick-borne diseases using Boolean searching (S1 Table). The inclusion criteria include studies that assess knowledge, attitudes, practices, or behaviors related to mosquito- or tick-borne diseases and are conducted in the US, regardless of study design. Non-empirical studies, such as protocol papers, systematic reviews, and scoping reviews, will be excluded.

This review will assess current practices regarding the utilization of KAP surveys, the themes addressed, and the way questions are formulated. The research team will contact the authors to obtain survey tools when KAP surveys are not readily accessible within the manuscript. The following data will be retrieved from the KAP surveys using a data extraction form: the first author's name, publication year and title, diseases, vectors, region and country, study design and sample size, how the survey/tool/scale was developed, and sources referenced, how the survey/tool/scale was validated, and the number of items included. Further solicitation of survey tools will be made from applied public health departments using personal networks to identify potentially unpublished tools. Items will be extracted and included in a master database that will be used to determine potential items for the draft modules.

Subsequently, public-facing materials such as guidelines, posters, and fact sheets will be collected from selected state health departments and the Centers for Disease Control and Prevention and World Health Organization websites. State health departments will be selected based on the severity of the mosquito-borne and tick-borne diseases, as indicated by the incidence rates and reported cases in their health department jurisdiction. Relevant data will be collected according to a predefined data extraction format, which includes reference, source, format (e.g., poster, fact sheet), vector, disease, domain (Knowledge, Attitude, or Practice), and sub-domain (e.g., symptom, transmission).

**Development of the draft survey tools.** The data systematically collected from the scoping review of peer-reviewed papers on KAP related to VBDs, along with prevention and control guidelines, will serve as the foundation for developing the modules. Once the initial item pools are generated, the team will conduct a thorough review to check clarity, relevance, and redundancy across items within each domain. Item wording and content will first be reviewed in a Word document format to facilitate detailed edits and group discussion, and subsequently reviewed in the survey platform to ensure formatting, readability, and user experience. Items that are unclear, redundant, or do not align well with the intended constructs will be revised or removed. A consensus-based process will be used to refine the items. In cases where disagreements arise among the team, an external expert with relevant domain knowledge will be consulted to assist in resolving differences.

**Underlying theoretical frameworks.** As the goal of KAP surveys for VBD prevention and control is to identify areas to focus efforts that motivate action, survey items will be mapped to the components of two theoretical frameworks for health behavior, the Health Belief Model (HBM) and the Risks, Attitudes, Norms, Abilities and Self-Regulation framework (RANAS) [6,7]. Employing the HBM as a theoretical framework in survey development can be beneficial. This model helps gauge individual perceptions of risk severity as well as self-efficacy, or an individual's belief in their ability to take meaningful action to mitigate risk [6]. Given that risk perception is intrinsically aligned with motivating behavior change, these elements are essential in gauging the likelihood of success of an intervention. The RANAS framework complements

the HBM by addressing societal norms, which are not addressed in the HBM, thus emphasizing the importance of tailoring and contextualizing specific behaviors/behavior changes within a population [7].

### Phase two: Content validity through the Delphi

During phase two, content validity will be measured through the Delphi process. A group of expert judges will evaluate each item for content relevance, representativeness, and technical quality [10]. These expert judges will be comprised of representatives from multiple groups including:

1. Vector control personnel

2. Health department personnel

3. Academic VBD researchers

Expert opinions will be solicited via email. The initial recruitment will be among 30–50 individuals with an anticipated response of 70%. Those who agree to participate will engage in two rounds as part of the Delphi process. Strategies to keep participants engaged will be employed, including personalized survey invitations and periodic reminders. Three follow-up reminders will be sent every two weeks following the initial distribution. One phone call reminder will be conducted as needed for the second round to ensure a 70% follow-up for round two.

During round one, the expert panel will receive a link to a web-based Delphi questionnaire, which will include a consent form, background information, instructions for the survey, a short demographic survey, and the modules. A content validity ratio (CVR) will be employed as a measurement consensus for items. Each item will be scored using a 4-point scale: 4 = Highly Relevant, 3 = Highly Relevant But Needs Rewording, 2 = Somewhat Relevant, 1 = Not Relevant, and the CVR will be calculated [11]. The members of the panel will be able to suggest rephrasing, provide rationales for their choices through a comment box attached to each item, and suggest missing or new items. This study will involve at least 20 experts, so a minimum CVR value of 0.42 will be used [12]. Items falling below this threshold, indicating high disagreement among expert judges, will be removed.

Responses from round one will be analyzed and summarized into one feedback report, which will include summary statistics including the standard deviation and means for each item, the level of consensus, coefficient variation, participants' feedback and comments, and the decision regarding item selection based on predefined consensus level using the CVR. The feedback report will be sent back to the expert panel with the introductory material for the second round. During round two, experts will be asked to review the feedback report and re-evaluate items. Once a consensus has been met for all items, results will be sent out to the panel.

### Phase three: Face validity through cognitive interviews and survey

During phase three, cognitive interviews and surveys with target audience groups will be conducted to ensure the face validity of the modules. End users will judge whether the items are appropriate to the targeted construct and assessment objectives [10]. Given the target of the survey modules are adults who reside in areas with mosquito-borne disease and tick-borne disease activity, participants will be drawn from a diverse range of educational backgrounds and geographic locations (i.e., urban, suburban, and rural).

**Cognitive interviews.** To assess the extent to which questions reflect the domain of interest and answers produce valid measurements, we will conduct cognitive interviews with the target population [13]. The cognitive interview process will use think-aloud and verbal probes to identify how participants process and respond to items within the surveys [13]. Cognitive interviews will assist in identifying areas of concern that need to be addressed prior to administering the survey to a broader population. This is similar to pre-testing the surveys, but the interviews are more rigorously designed to elicit detailed feedback. There will be two groups of interviewees: one group will review all mosquito modules (n = 20),

and one group will review all tick modules (n = 20). With the final target of the public residing in areas with mosquito and tick-borne disease risk in mind, interviewees will be chosen to represent urban and rural areas and varying educational levels. Residents living in urban versus rural regions may have different insights and experiences with mosquitoes and ticks, thus including participants from both geographic regions will increase representation and ideally result in differing perspectives on module content. Interviewees with experience conducting qualitative research will ask each survey question to the participant. After the interviewee answers the question, probing questions will be asked to obtain feedback and allow participants to verbalize their thought processes. Example probes will include, "can you explain what [key term] means in your own words?", "I noticed you hesitated, can you tell me more?", "did you find this question easy to answer?" and overall questions, "did you find any of these questions difficult to answer?", "which questions were hard to answer". Interviewers will also ask participants to score each survey item on a three-point scale ranging from –1–2. They will score items based on readability, understandability, and relevance. Numeric scores will allow for a quantitative analysis that can supplement the qualitative feedback. Items will be modified based on the feedback for improved clarity [10].

To minimize missing data, if individuals do not want to answer a specific question, follow up probing will be conducted to understand why, i.e., is it a sensitive topic, or is there lack of clarity around the intention of the question. If individuals do not complete the cognitive interview, a qualitative assessment of the responses provided will be made to inform overall direction and scope of the survey and feedback for the specific questions that were assessed by the participant. Additionally, a replacement participant will be identified to obtain a full sample size of 20 completed interviews.

**Survey administration.** Computer-assisted personal interviewing (CAPI) using an online survey tool will be conducted to ensure that the surveys can collect data with minimum measurement errors and discriminate between audiences who are knowledgeable and have good practices versus those who may not [10]. In-person interviews will also be conducted (n = 50) to delineate differences in how questions should be asked during in-person interviews when online tools are not used. Scale items will be evaluated with three sample groups that reflect and capture the range of the target population and ensure external validity. To ensure external validity, the survey must capture a diverse and representative sample that reflects real-world variations in KAP regarding VBDs. The survey participants will include three groups: 1) individuals involved in the control and prevention of VBDs (Group 1, n = 75), 2) public health personnel not affiliated with VBD programs (Group 2, n = 75), and the general public (Group 3, n = 800). The inclusion of three distinct groups – VBD professionals, public health personnel, and the general public – will enhance the study's generalizability by incorporating multiple levels of expertise and engagement and provide independent validation that our tools can discriminate among different levels of KAP. English-speaking adults 18 and older will be eligible to participate, excluding incarcerated individuals. All survey items will be written at a 5th grade reading level using the Fry Graph Calculator as a measurement tool [14]. At this time, validation is for English-speakers only, but the goal is to validate the modules in Spanish in the future.

Participants for Group 1 will be identified by extending invitations through collaborative networks such as the American Mosquito Control Association, Centers for Disease Control and Prevention Centers of Excellence for Vector-borne Diseases and Regional Training and Evaluation Centers (TECs) and other known networks. Group 2 will be recruited through public health organizations such as the National Association of County and City Health Officials (NACCHO), current author connections with public health agencies. Snowball sampling will be conducted for both Group 1 and 2, whereby individuals can refer others who should take the survey at the end of their survey. The investigative team will then invite those individuals. Group 3 will be recruited using a public survey panel. As one of the goals is to assess how knowledge, attitudes, and practices differ based on age, gender, residential area, and occupation recruitment, goals will be stratified by these variables.

Sample sizes per group will be conservatively determined based on comparisons between Group 1 and Group 2 given these are more specialized populations with a much smaller target population than the general population. We estimate that a meaningful difference in proportions is 20% or more between the vector control population and those in public

health practice. Sample size estimates for a one-sided difference in proportion (20% vs. 40%) with standard ($\alpha = 0.05$, $\beta = 0.80$) are 64 per group; this increases to 74 per group (30% vs. 50%). A wide range of differences in proportion is anticipated given the survey will intentionally include more common and more rare knowledge and practice items to facilitate discrimination between populations. The higher sample size in Group 3 (general public) will allow internal comparisons among demographic sub-groups including individuals from differing education and income levels. Given that surveys are subject to non-response bias, we will assess the generalizability of the findings by comparing the demographics of the respondents with those of the broader invited sample and the general population (age 18 and older). Comparisons will be made across age, gender, education level, and residential area using corresponding census data.

**Analysis plan.** Descriptive statistics will be computed for participant demographics including age, gender, educational attainment, and residential setting (urban or rural). The levels of knowledge, attitudes, and practices across demographic variables such as age groups, gender, occupation, and education levels will be analyzed to identify vulnerable populations with limited understanding and suboptimal practices. This analysis aims to detect statistically significant differences among these groups (Groups 1, 2, and 3). Missing data from survey responses will be handled as a data point – i.e., informing us that the question may need further examination in terms of clarity and relevance. Additionally, if the data are missing for the last set of questions, it may indicate the tool is too lengthy. If there are questions that have higher rates of missing data within the general public group as compared to the vector control or public health group, this may signal the need for revision of that specific question for a general audience. To determine if the survey tools can discriminate between groups with *a priori* knowledge differences, demographic differences will be assessed between responders/ non-responders. If the data are missing at random, listwise deletion will be conducted. If data are not missing at random, mean imputation will be used. Further analysis will be conducted as needed including Cronbach's alpha to assess the reliability and factor analysis to refine the survey by removing items with low loadings. The expected timeline is outlined in Table 1.

## Discussion

This manuscript describes a protocol for the development of standardized VBD survey modules that can be used throughout the US to generate data that are comparable across diverse regions. Based on our literature review, VBD KAP surveys are widely used in the field, but they are inconsistent in terms of structure and implementation. Two of the most pressing issues are inconsistencies with the structure and wording of questions, and the reliability and validity of questions. Among the reviewed surveys, the classification of knowledge, attitudes and practices questions was not uniformly applied across all studies. For example, items listed in the practices section of one survey were listed in the attitudes section of another survey, and vice versa. This highlights the importance of using theoretical frameworks to guide the design of the survey. There were also instances when different research teams were trying to obtain the same information, but because their survey questions were asked differently, they were not comparable. Key areas in which questions differed included:

• Use of dichotomous vs. Likert scales for frequency or intensity of actions, perceptions and/or knowledge. Likert scales also differed on the number of options ranging from 3–7, making the responses incomparable across surveys.

**Table 1. Expected timeline.**

| Phase | | Contents | Timeline |
|---|---|---|---|
| Phase 1 | Domain Identification and Item Generation | Item creation and draft tool development | April-May 2025 |
| Phase 2 | Content Validity through the Delphi | Round 1 Delphi for item selection and modification | June-July 2025 |
| | | Round 2 Delphi for item selection and modification | August-September 2025 |
| Phase 3 | Face Validity through Cognitive Interviews and Survey | Cognitive interviews | October-December 2025 |
| | | Survey administration | January-April 2026 |

- Time scales of reference (seasonal vs. past week vs. past year, vs. ever).

- Inclusion or exclusion of additional context in inquiring about actions (i.e., no context vs. when you are outside vs. when you are being bitten by mosquitoes).

- Differences in pre-text – provision of additional information about the next questions.

- Order of questioning, though many proceeded from knowledge to attitudes, to practices.

- Use of a behavioral theory as an underlying theoretical framework.

Furthermore, it was often unclear how research teams developed their surveys because references, theoretical frameworks, and sources for question derivation were not provided. Findings differed based on how questions were asked and the answer choices that were allocated, impeding the comparability of results across different surveys. The research community is recognizing the importance of this issue. A recent survey tool was developed for Europe, MosquitoWise, that has taken the first steps to create a validated tool for assessing KAP surveys for mosquito-borne diseases [15]. This tool has already received significant use within the past year demonstrating the important need in this area. However, the MosquitoWise survey has several limitations. It is framed for the European context, had limited input from vector control and prevention experts, does not address tickborne diseases, and used only four expert reviewers and five cognitive interviews [15]. Our proposed work increases the rigor of the validation process by including a broader set of experts from applied and academic backgrounds, using a modular framework that will allow investigators to include both core mosquito and tickborne disease questions, and to enhance those with questions specific to the disease systems under investigation.

The resulting survey modules developed using this protocol will guide public health professionals and researchers in the development of future KAP studies. Open access to the developed survey items and modules can maximize uniformity in KAP survey design and structure across geographical regions and enhance comparability over time and geography. Establishing a three-phase process that incorporates a comprehensive literature review, a rigorous process for content validation, and testing for face validity will result in survey modules that can be broadly disseminated for use across the US. This is particularly important in areas where vectors and/or pathogens are newly emerging so that patterns of disease transmission can be compared across geographic regions. The surveys developed via this protocol will be available for use by research teams that are interested in better understanding the knowledge, attitudes, and practices of the general public regarding VBDs. Public health, vector control agencies, and applied research programs can leverage these validated survey items to conduct a variety of assessments. Example uses include the assessment of tailored educational campaigns such as the deployment of federal or state programming at the local level. Using validated items will assist in consistent identification of program success. Similarly, these survey items could be used to determine the impact of programming on targeting high risk communities. Finally, when implemented consistently over time, leveraging a validated survey tool can determine changing behavioral patterns related to vector prevention and control. This protocol process does not include survey development for Spanish-speaking populations. However, after the initial modules have been finalized, future development will expand to include these populations. Creating tools that rigorously assess community KAP will aid in the development and implementation of practical and sustainable interventions to reduce the burden of disease.

## Supporting information

**S1 Table. Search criteria used during the literature review.**
(DOCX)

## Acknowledgments

We wish to thank Joshua Arnbrister for his support during the early stages of this project.

## Author contributions

**Conceptualization:** Mary H. Hayden, Kacey Ernst.

**Funding acquisition:** Kacey Ernst.

**Investigation:** Skyler Finucane, Kacey Ernst, Sarah Yeo.

**Methodology:** Mary H. Hayden, Kacey Ernst, Sarah Yeo.

**Supervision:** Kacey Ernst.

**Writing – original draft:** Skyler Finucane, Mary H. Hayden, Kacey Ernst, Sarah Yeo.

**Writing – review & editing:** Skyler Finucane, Alexandria Renault, Mary H. Hayden, Kacey Ernst, Sarah Yeo.

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
