## [Decision Letter · Decision Letter 0]

21 May 2025

Dear Dr. Ernst,

Thank you for submitting your manuscript to PLOS ONE. After careful consideration, we feel that it has merit but does not fully meet PLOS ONE’s publication criteria as it currently stands. Therefore, we invite you to submit a revised version of the manuscript that addresses the points raised during the review process.

We look forward to receiving your revised manuscript.

Kind regards,

Maria Stefania Latrofa

Academic Editor

PLOS ONE

Reviewers' comments:

Reviewer's Responses to Questions

**Comments to the Author**

1. Does the manuscript provide a valid rationale for the proposed study, with clearly identified and justified research questions?

Reviewer #1: Yes

2. Is the protocol technically sound and planned in a manner that will lead to a meaningful outcome and allow testing the stated hypotheses?

Reviewer #1: Yes

3. Is the methodology feasible and described in sufficient detail to allow the work to be replicable?

Reviewer #1: Yes

4. Have the authors described where all data underlying the findings will be made available when the study is complete?

Reviewer #1: Yes

5. Is the manuscript presented in an intelligible fashion and written in standard English?

Reviewer #1: Yes

You may also provide optional suggestions and comments to authors that they might find helpful in planning their study.

Reviewer #1: The manuscript PONE-D-25-17579, “Developing and validating modular surveys for vector-borne diseases: a study protocol”, presents a well-structured and timely protocol for the development and validation of standardized Knowledge, Attitudes, and Practices (KAP) surveys targeting vector-borne diseases (VBDs) in the United States. Given the increasing burden of VBDs and the current lack of harmonized tools for assessing public knowledge and behavior, this protocol addresses a significant methodological gap. The use of theoretical frameworks (HBM and RANAS), a phased development process, and diverse target populations adds to its scientific rigor. The manuscript is suitable for publication in PLOS ONE pending minor revisions related to writing clarity and a few enhancements to methodological transparency. Overall, it offers a valuable contribution to public health research and surveillance strategies.

- There’s no mention of how missing data from cognitive interviews or surveys will be addressed.

- It’s unclear how many initial items per domain are expected to be tested, and how item reduction will be handled. Please add information, such as: "Initial item pools will intentionally include overlapping items within each domain to allow for assessment of internal consistency and clarity. Item reduction will be based.... and items that do not meet these criteria will be considered for removal or rewording.”

- There’s a well-described sampling plan, but no mention of how nonresponse bias will be handled. Add information on how you’ll assess or mitigate nonresponse bias, such as: “To assess potential nonresponse bias, we will compare basic demographic characteristics (e.g., age, gender, education level, residential area) between survey respondents and the broader invited sample, where such data are available. This will help identify systematic differences that may affect generalizability.”

- The discussion mentions that survey results can inform interventions but doesn’t explain how results will be used by agencies. Suggestion - Add a paragraph in the discussion proposing potential use cases, such as: tailoring educational campaigns, targeting high-risk communities, evaluating longitudinal behavior change following. interventions.

**Do you want your identity to be public for this peer review?** For information about this choice, including consent withdrawal, please see our Privacy Policy

Reviewer #1: No

---

## [Author Response · Author response to Decision Letter 1]

14 Jun 2025

All comments have been addressed in the response to reviewers file attachment.

---

## [Decision Letter · Decision Letter 1]

21 Jul 2025

Developing and validating modular surveys for vector-borne diseases: a study protocol

PONE-D-25-17579R1

Dear Dr. Ernst,

We’re pleased to inform you that your manuscript has been judged scientifically suitable for publication and will be formally accepted for publication once it meets all outstanding technical requirements.

Kind regards,

Maria Stefania Latrofa

Academic Editor

PLOS ONE

Reviewers' comments:

Reviewer's Responses to Questions

**Comments to the Author**

1. Does the manuscript provide a valid rationale for the proposed study, with clearly identified and justified research questions?

Reviewer #1: Yes

2. Is the protocol technically sound and planned in a manner that will lead to a meaningful outcome and allow testing the stated hypotheses?

Reviewer #1: Yes

3. Is the methodology feasible and described in sufficient detail to allow the work to be replicable?

Reviewer #1: Yes

4. Have the authors described where all data underlying the findings will be made available when the study is complete?

Reviewer #1: Yes

5. Is the manuscript presented in an intelligible fashion and written in standard English?

Reviewer #1: Yes

You may also provide optional suggestions and comments to authors that they might find helpful in planning their study.

Reviewer #1: All suggestions have been addressed, and the manuscript has significantly improved. I recommend the publication of the manuscript in its current form.

**Do you want your identity to be public for this peer review?** For information about this choice, including consent withdrawal, please see our Privacy Policy

Reviewer #1: No

---

## [Editor Report · Acceptance letter]

PONE-D-25-17579R1

PLOS ONE

Dear Dr. Ernst,

I'm pleased to inform you that your manuscript has been deemed suitable for publication in PLOS ONE. Congratulations! Your manuscript is now being handed over to our production team.

Kind regards,

on behalf of

Dr. Maria Stefania Latrofa

Academic Editor

PLOS ONE